# Identification of Key Modules and Candidate Genes for Powdery Mildew Resistance of Wheat-*Agropyron cristatum* Translocation Line WAT-2020-17-6 by WGCNA

**DOI:** 10.3390/plants12020335

**Published:** 2023-01-11

**Authors:** Mingming Yao, Xinhua Wang, Jiaohui Long, Shuangyu Bai, Yuanyuan Cui, Zhaoyi Wang, Caixia Liu, Fenglou Liu, Zhangjun Wang, Qingfeng Li

**Affiliations:** 1School of Agriculture, Ningxia University, Yinchuan 750021, China; 2State Key Laboratory of Crop Gene Exploration and Utilization in Southwest China, College of Agronomy, Sichuan Agricultural University, Chengdu 611130, China

**Keywords:** wheat, *Agropyron cristatum*, powdery mildew, transcriptome, WGCNA

## Abstract

As one of the serious diseases of wheat, powdery mildew (*Blumeria graminis* f. sp. *tritici*) is a long-term threat to wheat production. Therefore, it is of great significance to explore new powdery mildew-resistant genes for breeding. The wild relative species of wheat provide gene resources for resistance to powdery mildew breeding. *Agropyron cristatum* (2n = 4x = 28, genomes PPPP) is an important wild relative of wheat, carrying excellent genes for high yield, disease resistance, and stress resistance, which can be used for wheat improvement. To understand the molecular mechanism of powdery mildew resistance in the wheat-*A. cristatum* translocation line WAT2020-17-6, transcriptome sequencing was performed, and the resistance genes were analyzed by weighted gene co-expression network analysis (WGCNA). In the results, 42,845 differentially expressed genes were identified and divided into 18 modules, of which six modules were highly correlated with powdery mildew resistance. Gene ontology (GO) enrichment analysis showed that the six interested modules related to powdery mildew resistance were significantly enriched in N-methyltransferase activity, autophagy, mRNA splicing via spliceosome, chloroplast envelope, and AMP binding. The candidate hub genes of the interested modules were further identified, and their regulatory relationships were analyzed based on co-expression data. The temporal expression pattern of the 12 hub genes was verified within 96 h after powdery mildew inoculation by RT-PCR assay. In this study, we preliminarily explained the resistance mechanism of the wheat-*A. cristatum* translocation lines and obtained the hub candidate genes, which laid a foundation in the exploration of resistance genes in *A*. *cristatum* for powdery mildew-resistant breeding in wheat.

## 1. Introduction

Common wheat (*Triticum aestivum* L., 2n = 6x = 42, AABBDD) is one of the staple crops in the world and plays an important role in food security [1,2,3]. Wheat powdery mildew (*Blumeria graminis* f. sp. *tritici*, *Bgt*) is one of the most threatening diseases of wheat [4]. The infection of wheat powdery mildew led to an incipient wilting of leaves and a decrease in tiller number, spike rate, grain number per spike, and 1000-grain weight, which seriously affected the yield and quality [5]. In order to reduce the impact of powdery mildew, fungicides were usually used in wheat production, which not only led to the increase of drug resistance in pathogens but also caused environmental pollution and ecological destruction [6]. Comparatively, breeding powdery mildew resistant varieties is considered the most effective and environmentally friendly way to prevent disease epidemics [7]. Since most of the resistance genes often break down once they are widely deployed in wheat cultivars due to the emergence of new *Bgt* isolates, it is important to explore new genes with broad-spectrum resistance. Wild relatives of wheat are important resources and contain a large number of powdery mildew resistance genes that can be used for wheat improvement. There were lots of powdery mildew-resistant genes identified from wheat wild relatives, such as *Pm2b* [8], *Pm8* [9], *Pm17* [10], *Pm21* [11], *Pm29* [12], *Pm34* [13], *Pm35* [14], *Pm40* [15], *Pm43* [16], *Pm51* [17], *Pm55* [18], *Pm56* [19], *Pm58* [20], *Pm62* [21], *Pm66* [22], and *Pm67* [23], among which *Pm21* from *Dasypyrum villosum* was most widely used in wheat genetic improvement.

*Agropyron cristatum* (L.) Gaertn. is one of the main wild relatives of wheat with high-yield characteristics, such as multiple tillers, spikelets, and florets [24], and multiple disease-resistant traits, including wheat powdery mildew, yellow dwarf, and rust [25]. In order to introduce the beneficial genes from *A. cristatum* into wheat, a large number of genetic resources containing the desired genes have been created, including wheat-*A. cristatum* 6P, 1P addition lines [26,27], 6P, 2P translocation lines [28,29,30,31,32,33,34], and introgression lines [35,36]. In addition, new wheat varieties containing genes for the excellent characteristics of *A. cristatum* were successfully developed, such as Pubing 143, Jinmai 80, Pubing 9946, Pubing 701, Xindong 49 (Pubing 696), Kenong 2011, Pubing 151, Qingmai 8, Chuanmai 93, and Pubing 2011 [37]. Previously, we identified several wheat germplasm resources with powdery mildew-resistant genes, such as the wheat-*A. cristatum* 2P addition line, 6P addition line, and translocation line. The above materials laid a foundation to explore resistance genes and study their mechanism.

With the rapid development of biotechnology, high-throughput sequencing contributes to systematically study complex biological problems and reveals their molecular mechanisms. Transcriptome sequencing technology could provide almost all genetic information of a species in a certain state quickly and comprehensively, making it an important tool for studying the level of gene transcription and identifying functional genes. Weighted gene co-expression network analysis (WGCNA), an efficient method for studying correlation networks, could cluster genes into modules according to their expression patterns [38,39]. It is important that hub genes could be identified and explored based on the relationship between the modules and traits [40,41]. For example, 3 modules and 12 candidate genes related to drought resistance in wheat at different growth stages were obtained through WGCNA. What’s more, Zheng et al. analyzed the genes and their expression patterns related to epidermal waxy pathway using WGNCA and using the transcriptome data of maize, both wild-type and mutant [42]. In this study, we conducted transcriptome sequencing of disease-resistant and susceptible materials under dynamic stress of powdery mildew, and we utilized WGNCA to identify gene modules and hub genes associated with the process of powdery mildew stress in wheat. The results provided a theoretical basis for further understanding the molecular mechanisms of powdery mildew stress and laid a foundation for exploring alien desirable genes in the progeny of wheat distant hybridization.

## 2. Results

### 2.1. Identification of Disease Resistance and Cytology

The results of powdery mildew resistance identification in the field showed that the material WAT-2020-17-6 was resistant and the material W-2020-17-1 was susceptible (Figure 1A). According to the results from cytological analysis and genomic in situ hybridization (GISH) detection, WAT-2020-17-6 is a wheat-A. cristatum homozygous 6P translocation line (Figure 1B), while W-2020-17-1 is a non-translocation line (Figure 1C).

### 2.2. Data Collection and Preprocessing

A total of 48 samples of two test materials inoculated with powdery mildew at eight time points were sequenced, and the raw data were filtered. A total of 3,399,214,794 clean reads and 509.9G clean bases were obtained from the transcriptome data. The percentage of Q20 and Q30 bases in each sample was 97.44–98.29% and 92.95–94.87%, respectively, and the GC content was 51.48–53.46%, indicating a high quality of the sequencing data. Matching data were obtained by comparing all quality control data to the reference genome of Chinese Spring, and the proportion of mapped reads in all samples exceeded 84.9%, indicating that the selection of the reference genome was appropriate (Appendix A).

### 2.3. Weighted Gene Co-Expression Network Construction

After screening and filtering the gene set, a total of 42,845 genes were obtained for constructing the weighted gene co-expression network. The selection of a soft threshold (Power) is the key step to constructing the network. When the soft threshold was set as 11 with a scale-free network fitting index of R^2^ > 0.80, average connectivity was close to 0 (Figure 2).

The required scale-free network could be obtained by choosing the appropriate soft threshold. According to the determined soft threshold (β = 11), the network was constructed, and genes were hierarchically clustered using dissimilarity to establish a hierarchical clustering tree (Figure 3A). Next, the tree was cut into different modules by the dynamic shear method, and modules with a correlation coefficient greater than 0.75 or a dissimilarity coefficient less than 0.25 were merged. Finally, 18 co-expression modules were obtained, represented by different colors (Figure 3A). The number of genes in each module is shown in Figure 3B. Among them, the Turquoise module (10,980) had the most genes, and the Grey60 module (75) had the least. The genes in the Grey module were not assigned to any module.

### 2.4. Correlation Analysis to Obtain the Interested Modules

Correlation analysis showed similar gene expression in the same sample groups, and there were significant differences between the sample groups that were suitable for WGCNA. The modular eigenvalues (ME) and their correlation with traits were calculated to identify modules that were significantly associated with the interested traits [38]. In this study, six co-expression modules specifically related to the stage of powdery mildew stress were identified (|r| > 0.70, *p* < 0.05) (Figure 4). Among them, the Midnightblue module (|r| = 0.75, *p* = 0.03) was positively correlated with powdery mildew stress at 12 h; the Blue module (|r| = 0.91, *p* = 0.002) was positively correlated with powdery mildew stress at 24 h; the Brown module (|r| = 0.82, *p* = 0.01) was positively correlated with powdery mildew stress at 36 h; the Lightcyan module (|r| = 0.79, *p* = 0.02) and Turquoise module (|r| = 0.75, *p* = 0.03) were positively correlated with powdery mildew stress at 48 h. The Tan module (|r| = 0.96, *p* = 2 × 10^−4^) was positively correlated with powdery mildew stress at 96 h (Figure 4). In order to understand the potential role of each gene to powdery mildew resistance, genes in the six selected modules (Midnightblue, Blue, Brown, Lightcyan, Turquoise, and Tan modules) were further analyzed.

### 2.5. Enrichment Analysis of Genes in Interested Modules

Six modules (Midnightblue, Blue, Brown, Lightcyan, Turquoise, and Tan modules), all related to the stages of powdery mildew stress in wheat-*A. cristatum* translocation lines and mainly enriched in biological process (BP), cellular component (CC), and molecular function (MF), were selected for GO enrichment analysis. The genes in the Midnightblue module were enriched in 278 biological processes, 37 cytological components, and 93 molecular functions and were also significantly enriched in N-methyltransferase activity (GO:0008170), S-adenosylmethionine-dependent methyltransferase activity (GO:0008757), arginine N-methyltransferase activity (GO:0016273), and the jasmonic acid metabolism process (GO:0009694) (Figure 5A). The Midnightblue module enrichment results were due to molecular function (MF). The Blue module genes were enriched in 1893 biological processes, 362 cytological components, and 777 molecular functions and were also significantly enriched in autophagy (GO:0006914), endoplasmic reticulum membrane (GO:0005789), an endoplasmic reticulum sub-compartment (GO:0098827), and the nuclear outer membrane–endoplasmic reticulum membrane network (GO:0042175) (Figure 5B). The Blue module enrichment results were due to a biological process (BP) and cellular component (CC). The Brown module was enriched in 1819 biological processes, 355 cytological components, and 712 molecular functions and were also significantly enriched in mRNA splicing via spliceosome (GO:0000398), ubiquitin-like protein-specific protease activity (GO:0019783), thiol-dependent ubiquitin hydrolase activity (GO:0036459), and ubiquitin hydrolase activity (GO:0101005) (Figure 5C). The Brown module enrichment results were due to molecular function (MF). The Lightcyan module was enriched in 102 biological processes, 31 cytological components, and 40 molecular functions. The Lightcyan module was enriched in transcriptional regulator activity (GO:0003712), 4-hydroxy-tetrahydrodipicolinate reductase (GO:0008839), regulation of fatty acid metabolic process (GO:0019217), and NADPH binding (GO:0070402) (Figure 5D). The Lightcyan module enrichment results were due to molecular function (MF). The Turquoise module was enriched in 1972 biological processes, 365 cytological components, and 807 molecular functions and was significantly enriched in a chloroplast envelope (GO:0009941), chloroplast thylakoid membrane (GO:0009535), plastid thylakoid membrane (GO:0055035), and photosystem I (GO:0009522) (Figure 5E). The Turquoise module enrichment results were due to a biological process (BP) and cellular component (CC). The Tan module was enriched in 323 biological processes, 95 cytological components, and 104 molecular functions and significantly enriched AMP binding (GO:0016208), phosphoenolpyruvate carboxylase activity (GO:0008964), the trans-Golgi network membrane (GO:0032588), and a cytoplasmic vesicle part (GO:0044433) (Figure 5F). The Tan module enrichment results were due to a cellular component (CC) and molecular function (MF).

### 2.6. Identification of Hub Genes

The first 150 connecting genes of the interested modules were analyzed and visualized by Cytoscape 3.9.0. The Cyto Hubba plug-in was used to screen hub genes. According to the connectivity, the top two genes in each module were selected as hub genes and compared with Arabidopsis sequences to predict the function (Table 1). Twelve hub genes were selected from the Midnightblue module (Figure 6A), Blue module (Figure 6B), Brown module (Figure 6C), Lightcyan module (Figure 6D), Turquoise module (Figure 6E), and Tan module (Figure 6F). Functional prediction showed that most of these genes were related to stress. For instance, *TraesCS6A02G281300* in the Midnightblue module might belong to the ERF transcription factor family. Studies have shown that the ERF transcription factor family is mainly involved in environmental stress and the hormone stimulation process, such as low temperature stress, hypoxia stress, and pathogen infection process [43]. *TraesCS5D02G265500* in the Turquoise module might belong to the F-box protein family MAX2, and a number of the F-box protein family could induce stomatal closure to improve resistance to bacteria in Arabidopsis [44]. *TraesCS5B02G426500* in the Turquoise module might belong to the CCCH zinc finger protein family. The RR-TZF protein in the CCCH zinc finger protein family is unique to plants. Studies have shown that the RR-TZF protein regulates gene expression by mediating RNA degradation to affect plant growth and stress response [45]. *TraesCS5A02G332900* in the Tan module might belong to the MYB gene family. MYB transcription factors are involved in the regulation of lignin metabolism. Lignins play a role in keeping the mechanical strength of plant cell wall and increasing the hardness of stems, thereby resisting damage caused by the external environment [46]. 

### 2.7. Validation of Hub Genes by RT-PCR

In order to verify the correlation between the core genes and the stages of powdery mildew resistance, these 12 hub genes were verified by RT-PCR. Compared with its expression at 0 h, the expression of *TraesCS1A02G225200* was significantly different at 6 h, 12 h, 24 h, 36 h, 48 h, and 96 h in WAT-2020-17-6, and it was also significantly different at 6 h compared with its expression at 96 h in W-2020-17-1 (Figure 7). The expression of *TraesCS6A02G281300* was significantly different at 6 h compared with its expression at 96 h in WAT-2020-17-6 and at 6 h, 36 h, 48 h, 72 h, and 96 h in W-2020-17-1 (Figure 7). The expression of *novel.8020* was significantly different at 6 h, 12 h, 36 h, 48 h, 72 h, and 96 h in WAT-2020-17-6 and at 6 h, 12 h, 24 h, and 36 h in W-2020-17-1. *TraesCS6D02G008600* was significantly different at 6 h, 24 h, 48 h, 72 h, and 96 h in WAT-2020-17-6 and significantly different at 6 h, 12 h, 36 h, 48h, and 96 h in W-2020-17-1. The expression of *novel.10169* was significantly different at 6 h, 12 h, 24 h, and 48 h in WAT-2020-17-6 but not in W-2020-17-1; *TraesCS1B02G274700* was significantly different at 48 h in WAT-2020-17-6 but not in W-2020-17-1; *TraesCS5D02G450500* was significantly different at 6 h in WAT-2020-17-6 but not in W-2020-17-1; *TraesCS7D02G526100* was significantly different in WAT-2020-17-6 at 6 h and 72 h and was significantly different from 6 h to 96 h in W-2020-17-1; *TraesCS5D02G265500* was significantly different at 6 h and 72 h in WAT-2020-17-6 and at 6 h, 12 h, 24 h, 36 h, 48 h, and 96 h in W-2020-17-1; *TraesCS5B02G426500* was significantly different at 6 h, 24 h, and 72 h in WAT-2020-17-6 and at 6 h to 96 h in W-2020-17-1; *TraesCS5A02G332900* was significantly different in WAT-2020-17-6 at 24 h, 36 h, and 96 h and in W-2020-17-1 at 6 h, 12 h, 24 h, 48 h, 72 h, and 96 h; *novel.7975* was significantly different from 6 h to 96 h in comparison with that of 0 h in WAT-2020-17-6 and W-2020-17-1.

## 3. Discussion

The discovery, development, and utilization of excellent germplasm resources could contribute to crop genetic improvement [47]. Due to the extensive use of the same or similar parents, long-term directional selection of varieties, and the simplification of cultivated varieties, the genetic background of existing wheat varieties has become increasingly narrow [48], which limits the improvement of wheat yield and quality and weakens the resistance of biotic and abiotic stresses [49]. The abundant resistance genes existing in wheat relatives could be introduced by distant hybridization [50]. For instance, rye is the most widely used wheat and is successfully used in wheat distant hybridization. The resistance genes, such as *Yr9* and *Pm8* in the 1RS chromosome and lots of high-yield genes, have been explored and utilized to breed new wheat varieties [51,52]. Since the 1960s, with the promotion and planting of dwarf stalk varieties and the increase of nitrogen fertilizer application and planting density, powdery mildew has become one of the most serious diseases of wheat production in the world. Chemical control has achieved positive effects in wheat powdery mildew, but it wastes manpower and material and financial resources, especially causing environmental pollution. Therefore, breeding different varieties with powdery mildew-resistant varieties is the most economical and effective method [53]. A large number of important resistant sources have been cultivated by transferring exogenous chromatin from *Haynaldia villosa*, *Elytrigia repens*, *Agropyron cristatum*, *Aegilops*, *Roegneria kamoji*, *Leymus secalinus*, *Barley*, and *Psathyrostachys juncea* to wheat. For example, Zhang et al. identified a wheat-*Dasypyrum villosum* T5VS.5AL translocation line resistant to powdery mildew by introducing 5V chromosome [18]. Li et al. obtained a wheat-*A. cristatum* 2P translocation line resistant to powdery mildew using 2P disomic addition line II-9-3 by ^60^Co-γ to irradiation [25]. This indicates that wheat relatives carrying resistance genes are an effective way to expand wheat genetic resources. 

Previous studies have found that most of the photosynthetic pathways have a positive response to powdery mildew stress after infection. Han et al. found that chlorophyll content, photosynthetic rate, and soluble protein content decreased after inoculation with powdery mildew in wheat, and the reduction rate in susceptible varieties was higher than that of disease-resistant varieties [54]. In this study, transcriptome sequencing and a weighted gene co-expression network analysis (WGCNA) were used to mine the genes related to powdery mildew resistance in a wheat-*A. cristatum* translocation line. A total of 12 genes related to powdery mildew resistance were excavated. For example, *TraesCS6A02G281300* in the Midnightblue module and *TraesCS6D02G008600* in the Blue module both belong to the ERF transcription factor family. ERF responds to various biotic or abiotic stresses by regulating various signaling pathways, including abscisic acid (ABA), jasmonate (JA), ethylene, salicylic acid (SA), and other signaling pathways [55]. AP2/ERF transcription factors are proven to be relevant to plant disease resistance signal transduction. Rice *OsERF922* is induced by ABA and *M. oryzae*. Inhibition of this gene expression by RNAi can enhance the resistance of rice to *M. oryzae*. Meanwhile, the expression levels of PR, PAL, and other genes related to rice disease resistance were significantly increased, and the ABA content in plants was reduced. Overexpression of *OsERF922* led to the susceptibility to the disease, and ABA content in tissues also increased correspondingly, indicating that it could negatively regulate disease resistance by regulating ABA content [56]. In conclusion, genes in the ERF transcription factor family might defend against powdery mildew by modulating ABA signaling pathways in wheat. 

*TraesCS5D02G450500* in the Lightcyan module belongs to the pectin lyase-like superfamily protein (PL), which was initially detected in pathogenic microorganisms. Studies have shown that pathogenic microorganisms degrade plant cell walls by secreting PL to achieve the purpose of invasion, and this process also stimulates the immune response of plants [57]. Plant PL has a variety of biological functions, including participating in the physiological process of the interaction with pathogenic microorganisms and causing allergic reactions in plants. The mutation of PMR6 in *Arabidopsis* altered the cell wall components and increased the resistance of plants to powdery mildew [58]. *TraesCS5A02G332900* in the Tan module belongs to the MYB gene family. After pathogen infection in *Arabidopsis*, AtMYB30 could immediately send signals to regulate the accumulation of salicylic acid, thereby affecting cell death in plants [59]. As a signaling complex downstream of salicylic acid synthesis, MYB can bind to the promoter sequence of the disease-resistant protein DNA and participate in the activation of the disease-resistant gene and in the transcription activation [59]. In addition, many MYB transcription factors were activated in defense responses to jasmonic acid and signaling molecules, and they then regulate the expression of downstream disease-resistant genes to reduce pathogen-induced damage [60,61,62]. In conclusion, genes in the PL and MYB family can regulate the interaction with pathogens by mediating multiple pathways to participation in powdery mildew resistance.

## 4. Materials and Methods

### 4.1. Plant Materials

The wheat-*A. cristatum* 6P disomic addition line 5113 (2n = 2x = 44) was generated by the distant hybridization of Z559 and Fukuho [63]. The common wheat cultivar Zhongzuo 9504 was used as a susceptible control in the powdery mildew assessment. The test materials WAT-2020-17-6 and W-2020-17-1 were the self-crossing progeny materials of the disomic additional line 5113. All the plant materials were provided by the Key Laboratory of Modern Molecular Breeding for Dominant and Special Crops in Yinchuan, Ningxia, China.

### 4.2. Chromosome Preparation

To distinguish between Fukuho and *Agropyron cristatum* chromosomes, GISH was carried out to determine the karyotype of WAT-2020-17-6 and W-2020-17-1. Chromosome spreads were prepared from root tip cells as described by Han [64]. The detailed procedure was as follows: The seeds of the tested materials were placed in a petri dish with wet filter paper and cultured in a constant temperature light incubator at 22–24 °C for about 24 h. After seed germination, the culture dish was transferred to a refrigerator at 4 °C for 24 h. Next, the root tip was transferred to a 22–24 °C incubator for 2 d, and the root tip was collected when the length of the root reached 1–2 cm. The root tips were treated with an ice-water mixture for 24 h, and the Carnot fixation solution I (anhydrous ethanol:glacial acetic acid = 3:1) was fixed for 2 d and placed at 4 °C. The apical meristems of the root tips were dissected and then digested in 10 μL of mixed enzyme solution containing 2% pectinase and 2% cellulase in a water bath at 37 °C for 1 h. After enzymolysis, the apical meristem was crushed with a dissecting needle and centrifuged at low speed (3000 r/min) for 3 min. The top liquid was poured off, and 20 μL of 100% acetic acid was added and mixed by vortex. A total of 7 μL of the cell suspension was dropped onto a slide. The GISH signals were observed via an Olympus AX80 fluorescence microscope (Olympus Corporation, Tokyo, Japan), imaged with a charge-coupled device (CCD) camera (Diagnostic Institute, Inc., Sterling Height, MI, USA), and modified with Photoshop CS6.

### 4.3. Infection Experiment, RNA Extraction, and Sequencing

A mixture of the current Bgt isolates from different areas of the Ningxia province was used to evaluate the powdery mildew resistance of all materials. The responses of adult plants to the disease were recorded according to grades 0–9, in which grades 0–4 were resistant and grades 5–9 were sensitive [65]. The mixed strains from Zhongzuo 9504 were used to inoculate the seedlings at the three-leaf stage, and the disease resistance of the materials was evaluated. Infection types (IT) were scored on 0–4 scale, with 0 as immune (no visible symptoms), 0_i_ as nearly immune (necrotic flecks), 1 as highly resistant (necrosis with low sporulation), 2 as moderately resistant (necrosis with medium sporulation), 3 as susceptible (no necrosis with medium to high sporulation), and 4 as highly susceptible (no necrosis with full sporulation) [66]. The powdery mildew was cultured in Zhongzuo 9504, and when the powdery mildew was fully grown in Zhongzuo 9504, the spores of the fungus were transferred to the test material of triloba single-stage. The test materials were grown in a light incubator at 28 °C (16 h light, 8 h dark). At the three-leaf stage, RNAs were isolated at 0 h, 6 h, 12 h, 24 h, 36 h, 48 h, 72 h, and 96 h before and after Bgt inoculation. Leaf RNA was extracted using RNAiso Plus (Total RNA extraction reagent) (TaKaRa). The samples were sent to Novogene Bioinformation Technology Co., Ltd. (Beijing, China) for sequencing. The mRNA with a polyA tail was enriched by Oligo (dT) magnetic beads, and the ribosomal RNA was removed from the total RNA. The first cDNA strand was synthesized in the M-MuLV reverse transcriptase system using fragmented mRNA as a template and random oligonucleotide as a primer; next, the RNA strand was decomposed by RNaseH, and the second cDNA strand was synthesized in DNA polymerase I system using dNTPs as raw material. The purified double-strand cDNA was end-repaired, an A-tail was added, and the sequencing joints were connected. The cDNA, about 250-300bp, was screened by AMPure XP beads for PCR amplification, the PCR products were purified by AMPure XP beads again, and libraries were finally obtained [67]. A total of 48 samples were subjected to transcriptome sequencing to obtain the gene expression data of each sample. Sequencing sample groups of WAT-2020-17-6 and W-2020-17-1 were set as VPC and VPK. The WAT-2020-17-6 treatment groups were named C20-0, C20-6, C20-12, C20-24, C20-36, C20-48, C20-72, and C20-96. The W-2020-17-1 treatment groups were named K21-0, K21-6, K21-12, K21-24, K21-36, K21-48, K21-72, and K21-96.

### 4.4. Data Collection and Preprocessing

FastQC [68] software (version 0.11.9) was used to evaluate the quality of the original sequencing data, followed by Fastp software (version 0.23.0) to remove joints and filter low-quality reads. Quality control and filtering of 48 transcriptome sequencing raw data were performed. The Reads with adapter (Adapter), the Reads containing N (N means that the base information cannot be determined), and the low-quality Reads (The number of bases with Q_phred_ ≤ 20 accounts for more than 50% of the entire Read length) were removed. In order to obtain the location information of Reads on the reference genome, HISAT2.0.5 was used to compare the filtered Clean Reads with the reference genome of Chinese Spring [69]. Based on the alignment results, the expression of all transcripts were calculated using StringTie1.3.3b. DESeq2 1.16.1 was used to analyze the differential expression between sample groups and set the threshold of screening differential gene |log2FC| > 1 and FDR ≤ 0.05 [70,71].

### 4.5. Weighted Gene Co-Expression Network Construction

After the read counts of the data were obtained using the feature Counts [72], the TPM value of gene expression was calculated by R software (R version 4.1.2). At least one set of 48 gene samples with TPM values greater than 10 was selected to construct the expression matrix and analyze the gene co-expression network. The WGCNA of all the genes was conducted using the R WGCNA package [73], and the threshold was determined using the scale-free network principle. In order to elucidate the function and mechanism of the gene in wheat powdery mildew resistance, WGCNAR software package (V1.51) was used to construct the co-expression network of this gene. Genes were divided into different modules based on their expression pattern. The gene–module correlation was analyzed using the Pearson correlation. Hub genes in the modules of interest were determined using the Cytohubba pluggin [74] in Cytoscape v3.9.1.

### 4.6. Screening and Functional Analysis of Hub Genes

To further explore the function of specific modules, a gene ontology (GO) function analysis of specific module genes was performed. Differentially expressed genes between samples were identified using R-pack edgeR v3.18 [75]. The genes with FDR less than 0.05 and absolute value |log2FC| > 1 were considered to be differentially expressed genes among different groups [76]. The differential genes obtained from transcriptome in each group were analyzed, and Blast2GO (https://www.blast2go.com/ (accessed on 26 September 2022)) was used to retrieve the GO function that was significantly enriched. In order to obtain the hub genes in the specific modules, the Cytoscape 3.9.0 software was used to visualize the gene interaction network. According to the connectivity of the genes in the modules and the FPKM value of the genes in the transcriptome data, the genes with high connectivity and high expression were selected as the hub genes in the module. In these networks, each node represents one gene, and genes connecting both ends are usually considered to have the same biological function. A gene regulatory network diagram can accurately screen out the candidate genes that have a regulatory relationship with hub genes and predict unknown gene functions by using the functions of known genes.

### 4.7. Validation of Hub Genes by RT-PCR

In order to verify the expression of hub genes under powdery mildew stress, these 12 genes were verified by RT-PCR. Primer Premier5 (version 5.0) software was used to design the forward and reverse primers of each gene (Appendix A). β-actin was used as the internal reference gene and synthesized by Shanghai Sangon Biological Company. The cDNA reverse transcription was performed according to the instructions for the PrimeScript™ RT reagent Kit with gDNA Eraser (Perfect Real Time) (Takara) kit. Genious 2× SYBR Green Fast qPCR Mix was used for real-time quantitative PCR. Fluorescence information was collected using a Jena qTOWER^3^G PCR instrument, and each sample was repeated 3 times. The relative expression level was calculated using the 2^−△△ct^ method.

## 5. Conclusions

In this study, a wheat-*A. cristatum* 6P translocation line WAT-2020-17-6 was obtained and proved resistant to powdery mildew. By further transcriptome sequencing and weighted gene co-expression network analysis, the mechanism of powdery mildew resistance was preliminarily explored, and candidate genes were determined. It was shown that genes in the interested modules were enriched in signal pathways that were related to stress resistance. The hub genes mainly belong to the ERF transcription factor family, F-box/RNI-like superfamily protein, MYB gene family, etc. The results revealed that genes in transcriptional and post-transcriptional regulation synergistically function to resist the invasion of powdery mildew, which laid a foundation in the exploration of the *Agropyron cristatum*-resistant genes in wheat breeding.

## Figures and Tables

**Figure 1 plants-12-00335-f001:**
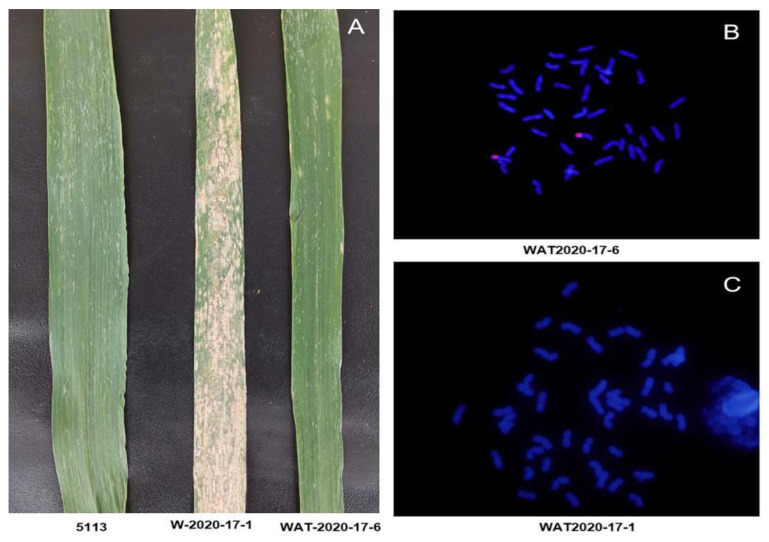
Evaluation of powdery mildew resistance and cytological identification of WAT-2020-17-6 and W-2020-17-1. (**A**) Disease responses to powdery mildew of 5113, WAT-2020-17-6 and W-2020-17-1. (**B**) GISH patterns showing WAT-2020-17-6, including a pair of translocation chromosomes. (**C**) GISH patterns showing W-2020-17-1, including 42 wheat chromosomes.

**Figure 2 plants-12-00335-f002:**
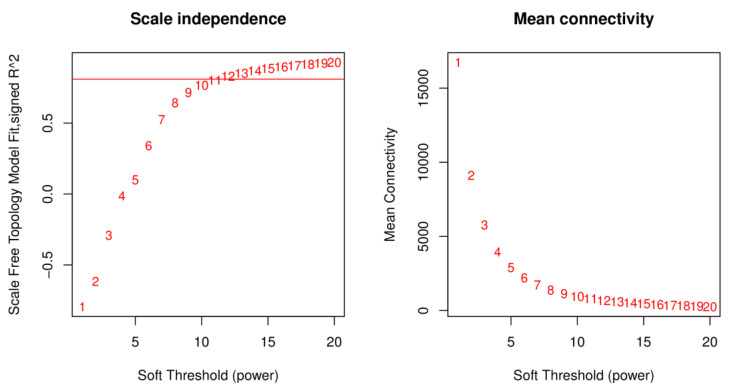
Determination of soft threshold. Network topology for different soft-thresholding powers of WAT-2020-17-6. The x-axis represents the soft threshold β. The y-axis of the left figure represents the square of the correlation coefficient between log(k) and log(p(k)) in the corresponding network. The y-axis of the right figure represents the mean of all gene adjacency functions in the corresponding gene module. The approximate scale-free topology can be attained at the soft thresholding power of 11.

**Figure 3 plants-12-00335-f003:**
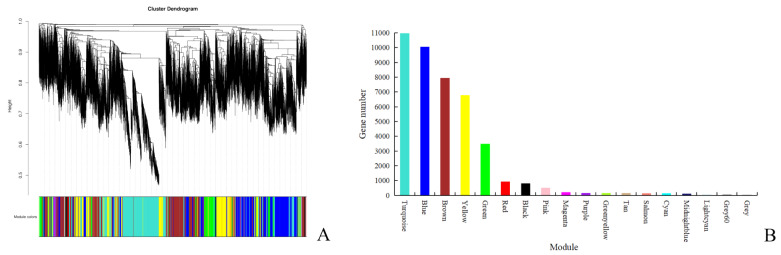
Clustering dendrograms of genes and module division. (**A**) Gene dendrogram by clustering the dissimilarity based on topological overlap. Each module was indicated by colors in the row, which contained a group of highly connected genes. (**B**) Distribution of gene number in modules. The x-axis represents different modules, and the y-axis represents gene numbers.

**Figure 4 plants-12-00335-f004:**
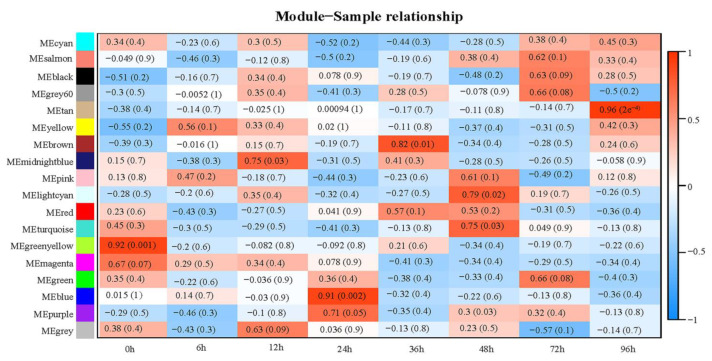
Correlation heatmap between modules and stage of powdery mildew stress. Each row in the table represents a module (indicated on the left), and each column represents a time point to powdery mildew stress (showing below). The table is colored according to the correlation (the *p* values are shown in parentheses), with a legend on the right.

**Figure 5 plants-12-00335-f005:**
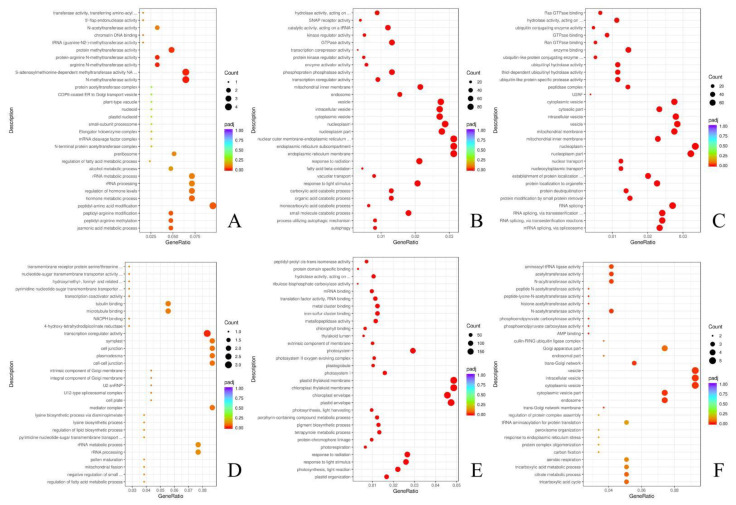
GO enrichment analysis of the interested modules related to powdery mildew stress. (**A**) Midnightblue module; (**B**) Blue module; (**C**) Brown module; (**D**) Lightcyan module; (**E**) Turquoise module; (**F**) Tan module. The size of the bubble represents the number of enriched genes. The Padj represents adjusted *p* value by a multiple hypothesis test, which is indicated by the bubble color with a legend on the right.

**Figure 6 plants-12-00335-f006:**
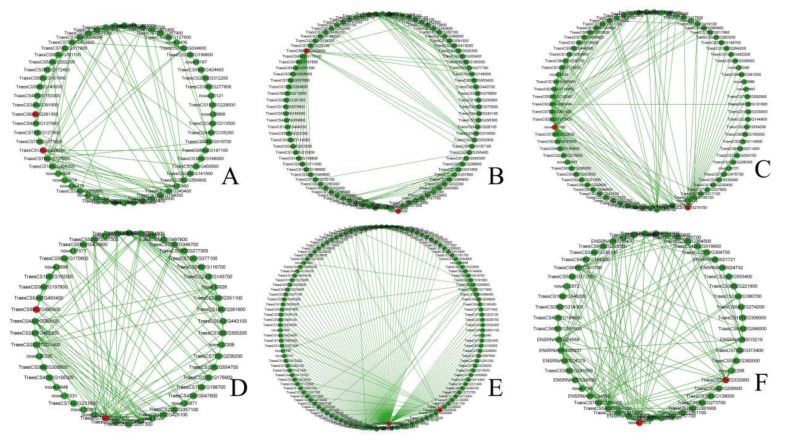
Identification of hub genes in the six interested modules. (**A**) The hub genes of the Midnightblue module are *TraesCS1A02G225200* and *TraesCS6A02G281300*. (**B**) The hub genes of the Blue module are *novel.8020* and *TraesCS6D02G008600*. (**C**) The hub genes of the Brown module are *novel.10169* and *TraesCS1B02G274700*. (**D**) The hub genes of the Lightcyan module are *TraesCS5D02G450500* and *TraesCS7D02G526100*. (**E**) The hub genes of the Turquoise module are *TraesCS5D02G265500* and *TraesCS5B02G426500*. (**F**) The hub genes of the Tan module are *TraesCS5A02G332900* and *novel.7975*. The top 150 linked genes of the module were analyzed and visualized, and the top 2 genes in each module were selected as hub genes according to the degree of gene connectivity.

**Figure 7 plants-12-00335-f007:**
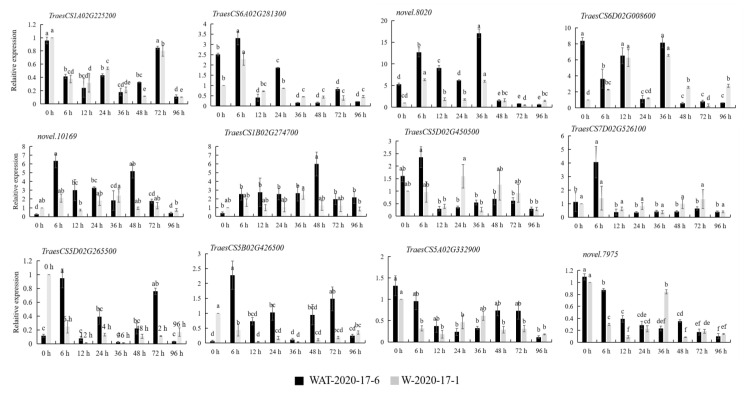
Relative expression levels of hub genes in materials WAT-2020-17-6 and W-2020-17-1. The x-axis represents the time points after inoculation with powdery mildew, and the y-axis represents the relative gene expression. The black column represents the material WAT-2020-17-6, and the gray column represents the test material W-2020-17-1. Different letters represented significant differences (*p* < 0.05).

**Table 1 plants-12-00335-t001:** Functional prediction of hub genes.

Module	Hub Gene ID	Gene ID in Aarbidopisis	Gene Function
Midnightblue	*TraesCS1A02G225200*	*AT3G11560.2*	LETM1-like protein
Midnightblue	*TraesCS6A02G281300*	*AT1G68550.1*	Encodes a member of the ERF (ethylene response factor) sub-family B-6 of ERF/AP2 transcription factor family
Blue	*novel.8020*	*AT3G02030.2*	Transferase
Blue	*TraesCS6D02G008600*	*AT1G50640.1*	Encodes a member of the ERF (ethylene response factor) sub-family B-1 of ERF/AP2 transcription factor family (ATERF-3)
Brown	*novel.10169*	*AT4G03205.2*	Coproporphyrinogen III oxidase
Brown	*TraesCS1B02G274700*	*AT5G63120.2*	P-loop containing nucleoside triphosphate hydrolases superfamily protein
Lightcyan	*TraesCS5D02G450500*	*AT4G13710.1*	Pectin lyase-like superfamily protein
Lightcyan	*TraesCS7D02G526100*	*AT1G62250.1*	Orotidine 5-phosphate decarboxylase
Turquoise	*TraesCS5D02G265500*	*AT4G13960.1*	F-box/RNI-like superfamily protein
Turquoise	*TraesCS5B02G426500*	*AT5G44260.1*	Encodes a Tandem CCCH Zinc Finger protein
Tan	*TraesCS5A02G332900*	*AT4G37180.2*	UIF1 is a nuclear and cytoplasmically localized myb-domain containing a member of the GARP G2-like sub-family of transcription factors.
Tan	*novel.7975*	*AT1G59840.1*	Co-factor assembly of complex C

## Data Availability

Transcriptome sequencing data are available from the NCBI under project ID PRJNA913840.

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
