# Peer review of "Identification of Key Modules and Candidate Genes for Powdery Mildew Resistance of Wheat-Agropyron cristatum Translocation Line WAT-2020-17-6 by WGCNA"

_plants, 2023, doi:10.3390/plants12020335_

Round 1
Reviewer 1 Report
The Agropyron cristatum is an important wild relative of wheat, which can be used for wheat improvement.
In this study, the authors sequenced wheat-Agropyron cristatum trans- 17 location line WAT2020-17-6 related 48 samples of transcriptomic under powdery mildew stress and analyzed the data, using WGCNA package to cluter the gene co-expression patterns and get some hub genes.
The dataset is valuable and the WGCNA results are interesting.
Comments:
1. The figure quality: the figures' size is too small and DPI is low, making them hardly recognize the details
2. Many software in the method section is not fully cited.
3. The raw data should upload to SRA or other databases.
4. Line 225: I think you presented the adjusted p-value not the original p-value in Figure 6. Make it clear in the legend.
5. Perhaps the authors could have focused on analyzing the results of WGCNA rather than illustrating the intermediate processes.
For example, Figure 2 is not necessary as the main image. Figure 3 and Figure 4 can be combined. Table 1 can be placed in the supplementary material.
Best
Author Response
Thank you very much for your advice. Please see attached for my reply.

Reviewer 2 Report
1-Add any figure under the related result.
2-Put the references based on the journal format.
Author Response

(The authors gave the same response as above.)

Reviewer 3 Report
In this study, the wheat-Agropyron cristatum WAT2020-17-6 by transcriptome sequencing, and identified 42,845 differentially expressed genes, and these differentially expressed genes were divided into 18 modules using WGCNA, of which 6 modules were highly correlated with powdery mildew stress. And author further identified candidate Hub genes associated with specific modules and analyzed their regulatory relationships based on co-expression data. In my opinion, this study needs some improvement.
1. 6 modules were highly correlated with powdery mildew stress, and author should do some experiments to verify.
2. Similarly, this part of the 12 Hub candidate genes related to powdery mildew resistance also needs some experimental verification.
3. Figures 5, 6, and 8 are very unclear and need to be replaced with new images.
4. Line 171-173, the author mentioned Figure 7, but I didn’t find it.
Author Response

(The authors gave the same response as above.)

Reviewer 4 Report
Please see attached for my comments.

Author Response

(The authors gave the same response as above.)

Round 2
Reviewer 1 Report
No more comment.
Author Response
Thank you very much for your advice and we would like to thank you again.
Reviewer 3 Report
The author has answered the questions I raised and this manuscript is recommended for acceptance.
Author Response

(The authors gave the same response as above.)

Reviewer 4 Report
All my comments are addressed. I do not have more comments.
Author Response

(The authors gave the same response as above.)
